# Pneumococcal genetic variability in age-dependent bacterial carriage

Philip HC Kremer[1]*, Bart Ferwerda[1,2], Hester J Bootsma[3], Nienke Y Rots[3], Alienke J Wijmenga-Monsuur[3], Elisabeth AM Sanders[3,4], Krzysztof Trzciński[4], Anne L Wyllie[4,5], Paul Turner[6,7], Arie van der Ende[8,9], Matthijs C Brouwer[1], Stephen D Bentley[10], Diederik van de Beek[1], John A Lees[11,12]*

[1]Department of Neurology, Amsterdam UMC, University of Amsterdam, Meibergdreef, Netherlands; [2]Department of Clinical Epidemiology, Biostatistics and Bioinformatics, University of Amsterdam, Amsterdam, Netherlands; [3]Centre for Infectious Disease Control, National Institute for Public Health and the Environment, Bilthoven, Netherlands; [4]Department of Pediatric Immunology and Infectious D, Wilhelmina Children's Hospital, Utrecht, Netherlands; [5]Epidemiology of Microbial Diseases, Yale School of Public Health, New Haven, United States; [6]Cambodia Oxford Medical Research Unit, Angkor Hospital for Children, Siem Reap, Cambodia; [7]Centre for Tropical Medicine and Global Health, Nuffield Department of Medicine, University of Oxford, Oxford, United Kingdom; [8]Department of Medical Microbiology and Infection Prevention, Amsterdam UMC, Amsterdam, Netherlands; [9]The Netherlands Reference Laboratory for Bacterial Meningitis, Amsterdam, Netherlands; [10]Parasites and Microbes, Wellcome Sanger Institute, Cambridge, United Kingdom; [11]European Molecular Biology Laboratory–European Bioinformatics Institute, Cambridge, United Kingdom; [12]MRC Centre for Global Infectious Disease Analysis, Department of Infectious Disease Epidemiology, Imperial College London, London, United Kingdom

*For correspondence:
philip_kremer@hotmail.com
(PHCK);
jlees@ebi.ac.uk (JAL)

Competing interest: The authors declare that no competing interests exist.

**Abstract** The characteristics of pneumococcal carriage vary between infants and adults. Host immune factors have been shown to contribute to these age-specific differences, but the role of pathogen sequence variation is currently less well-known. Identification of age-associated pathogen genetic factors could leadto improved vaccine formulations. We therefore performed genome sequencing in a large carriage cohort of children and adults and combined this with data from an existing age-stratified carriage study. We compiled a dictionary of pathogen genetic variation, including serotype, strain, sequence elements, single-nucleotide polymorphisms (SNPs), and clusters of orthologous genes (COGs) for each cohort – all of which were used in a genome-wide association with host age. Age-dependent colonization showed weak evidence of being heritable in the first cohort ($h^2$ = 0.10, 95% CI 0.00–0.69) and stronger evidence in the second cohort ($h^2$ = 0.56, 95% CI 0.23–0.87). We found that serotypes and genetic background (strain) explained a proportion of the heritability in the first cohort ($h^2_{serotype}$ = 0.07, 95% CI 0.04–0.14 and $h^2_{GPSC}$ = 0.06, 95% CI 0.03–0.13) and the second cohort ($h^2_{serotype}$ = 0.11, 95% CI 0.05–0.21 and $h^2_{GPSC}$ = 0.20, 95% CI 0.12–0.31). In a meta-analysis of these cohorts, we found one candidate association (p=1.2 × 10$^{-9}$) upstream of an accessory Sec-dependent serine-rich glycoprotein adhesin. Overall, while we did find a small effect of pathogen genome variation on pneumococcal carriage between child and adult hosts, this was variable between populations and does not appear to be caused by strong effects of individual genes. This supports proposals for adaptive future vaccination strategies that are primarily targeted at dominant circulating serotypes and tailored to the composition of the pathogen populations.

## Editor's evaluation

Strain variability in bacterial infections is a confounding factor in the treatment and prevention of the associated diseases. Pneumococcal disease is widespread, and the current vaccine targets only a subset of circulating strains, with disease and vaccine efficacy likely varying with the age of the host. Using two large databases of pneumococcal genomes, this study explores the associations between genomic factors and the age of the human host. Ultimately, these data and related studies will establish whether and how vaccines should be differentially designed for children and the elderly. This work will be of interest to those working in bacterial infections and host–pathogen genomics.

## Introduction

*Streptococcus pneumoniae* is a common commensal of the human upper respiratory tract and naso-pharynx, but can also cause pneumonia and invasive diseases such as sepsis or meningitis (*Bogaert et al., 2004*). Invasive pneumococcal disease (IPD) has a high mortality, and the overall mortality rate from IPD is higher in extreme age ranges, such as infants and the elderly (*Wahl et al., 2018*; *O Brien et al., 2009*). In the Netherlands, pneumococcal carriage rates are higher in children than in adults, with a prevalence of up to 80% at 2 years of age (*Wyllie et al., 2016*).

Pneumococcal carriage manifests as multiple carriage episodes of different serotypes. From birth, mucosal immunity builds up against different serotypes due to exposure, while immunity from maternal antibodies wanes. Host age is known to affect carriage prevalence and carriage duration of different serotypes (*Stearns et al., 2015*; *Turner et al., 2012b*), which is suggested to be driven by differences in immunity.(*Wyllie et al., 2020*) Studies in mice and humans showed evidence for age-dependent host–pathogen interactions involving interleukin (IL)-1 response in reaction to the pore-forming pneu-molysin (*ply*) toxin (*Kuipers et al., 2018*). IgA secretion is important in clearing *S. pneumoniae* from host upper respiratory tract mucosa, and this secretion is more effective in previously exposed individuals, the adults (*Binsker et al., 2020*). Bacterial genetics has been shown to explain over 60% of the variability in carriage duration, and specifically that the presence of a bacteriophage inserted in a mediator of genomic competence was associated with a decreased carriage duration (*Lees et al., 2017b*).

Pneumococci are highly genetically variable, displaying over 100 diverse capsular serotypes (*Ganaie et al., 2020*), which are a major antigen and the strongest predictor of carriage prevalence (*Croucher et al., 2018*). Pneumococcal conjugate vaccines (PCVs), targeting up to 13 capsule serotypes with high burden of invasive disease, decrease the rate of nasopharyngeal carriage and invasive disease (*Whitney et al., 2003*; *Poehling et al., 2006*). Besides a direct effect of vaccination with a PCV on the disease burden in the target population, that is, young children, it also reduces the disease burden caused by pneumococci with vaccine serotypes in the population not eligible for vaccination through indirect protection from colonization – reducing carriage rates in children reduces overall transmission of the most invasive serotypes (*Croucher et al., 2018*; *Desai et al., 2015*; *von Gottberg et al., 2014*). However, the introduction of PCV has resulted in the replacement of serotypes not covered by the vaccine (*Croucher et al., 2013*; *Corander et al., 2017*), which in some countries reaches levels of invasive disease return towards pre-vaccine levels (*Ladhani et al., 2018*; *Koelman et al., 2020*).

As not all serotypes can be included in a conjugate vaccine, three perspectives leading to improved pneumococcal vaccination have been proposed: whole-cell vaccines (*Malley et al., 2001*; *Campo et al., 2018*), protein vaccines (*Moffitt and Malley, 2016*), or changing components in the conjugate vaccine in response to the circulating population.(*Colijn et al., 2020*) Whole-cell vaccination trials are ongoing, but efficacy remains unproven in human populations (*Morais et al., 2019*). Protein vaccines contain antigens that elicit a strong mucosal immune response, with their targets chosen to be common or conserved in the target population, and ideally reducing onward transmission (*Pichichero, 2017*). In their current form, protein vaccines are not thought to be effective on their own, but if administered with serotype conjugates (possibly by replacing the carrier protein) they may help to reduce serotype replacement. Detailed modeling of the dynamics of pneumococcal population genetics has shown that targeting these vaccines towards serotypes prevalent in specific populations would likely be a superior strategy. This work further shows that providing age-specific vaccine design using complementary adult-administered vaccines (CAVs) is predicted to have the greatest effect on total IPD burden (*Colijn et al., 2020*). These authors also modeled including pilus in the vaccine, which

is more frequently present in a few key invasive serotypes and in infant carriage, but found this to be inferior to conjugate vaccines.

If proposing a future pneumococcal vaccination strategy based on host age, we should aim to better understand the differences between infant and adult carriage. Differences between other host niches have been found, some with a potential effect on onward transmission (*Lees et al., 2017c*; *Lees et al., 2017a*; *Zafar et al., 2019*). In particular, a previous study has suggested that the presence of pilus, which is found in a minority of pneumococcal isolates, has a selective advantage in infant carriage (*Binsker et al., 2020*).

We therefore wished to test three hypotheses. Firstly, carriage rates of individual strains or serotypes vary substantially between infants and adults in the same contact networks. Secondly, this variation is attributable at least in part due to pathogen genetic adaptation to either the infant or adult nasopharynx, which are immunologically different niches. Finally, this adaptation is due both to serotype and genetic background, and that some of the genetic effects are resolvable to individual genes, alleles, or regulatory variants that arise on multiple different genetic backgrounds due to a selective advantage. If we find clear associations, this would support proposals for age-specific vaccine design and may also suggest specific protein components that more broadly suppress carriage in the target age group than multivalent PCV alone.

For the first hypothesis, to test varying rates of carriage we can swab children and adults from the same population (and likely exposed to the same infection pressures) and quantify serotype and strain prevalence. For the second hypothesis, if we whole-genome sequence bacteria from these swabs, we can model the effects of all genomic variants on host age in a heritability analysis. This accounts for genetic variation that arose long ago and is fixed in particular lineage, but cannot map it to a particular region of the genome (*Earle et al., 2016*). To find individual genetic effects, which have necessarily occurred more recently and frequently, one study design would be to identify adult–infant transmission pairs and find variation that consistently occurs in localized regions of the genome, which would be particularly informative if also associated with a particular direction of transmission (*Lees et al., 2017c*). This removes genetic background as a confounder, giving a clean signal (*Young et al., 2012*). However, the identification of such pairs is very challenging for *S. pneumoniae*, and even when possible the small numbers limit power. We propose taking the more 'opportunistic' approach taken in genome-wide association studies (GWAS), where as many cases and controls (in this case, infant and adult samples) as possible are accumulated to boost statistical power, and genetic background is then controlled for in the association analysis. Where variation associated with age has arisen independently on multiple genetic backgrounds, GWAS has the ability to find these signals among the many lineage associations tested in the second step. In all cases, analysis can be improved by studying more than one population to determine whether findings are consistent among different host and pathogen populations.

To carry out these analyses, we used pneumococci isolated from nasopharyngeal swabs of 4320 infants and adults from the Netherlands (2009–2013) and Myanmar (October 2007–November 2008). Each cohort contains infant and adult samples from carriage, and there are significant differences between the host populations. This allows us to follow the above approach in each population and compare our findings between the populations. We present our findings with respect to each above hypothesis in turn and interpret them through the lens of using population genomics to determine optimal vaccination strategies.

## Results

We first analyzed the observed distribution of serotypes and strains in each of the two cohorts to assess overall trends of differences in carriage between adults and children exposed to similar forces of infection and look at the pathogen population's genetic heterogeneity between the two cohorts. Although our cohorts were broadly matched in the primary phenotype, age, large differences between the pathogen population are expected due to different geographies, social backgrounds, and only children in one cohort being vaccinated. Nucleotide variation across the entire genome can be used to cluster genetically related isolates into consistently named strains, called global pneumococcal sequence clusters (GPSC) (*Gladstone et al., 2019*). We used this over older gene-by-gene approaches such as MLST as it has been shown to represent biologically discrete clusters in the population, uses the full resolution available from whole-genome sequencing, and has strong community

support (*Gladstone et al., 2020*). For each sample, we enumerate the serotype (which is targeted by the vaccine) and GPSC membership, and count the number of each serotype observed in adult and child carriage.

## Serotypes and strains are variably carried between age groups and between cohorts

The Dutch cohort was made up of 1329 *S. pneumoniae* isolates comprising 41 unique serotypes (*Supplementary file 1*). Of these isolates, 689 (52%) comprised seven serotypes: 19A (225; 17%), 11A (111; 8%), 6C (97; 7%), 23B (84; 6%), 10A (67; 5%), 16F (54; 4%), and 21 (51; 4%). In this cohort of which the children were vaccinated, a minority of isolates (101; 8%) belonged to one of the vaccine serotypes (*Supplementary file 2*). A total of 3085 pneumococcal isolates of the Maela (unvaccinated) cohort comprised 64 unique serotype groups (*Supplementary file 3*). Of these isolates, 1631 (53%) comprised five serotypes: non-typable (511; 17%), 19F (402, 13%), 23F (307, 10%), 6B (236; 8%), and 14 (175; 6%). In the Dutch cohort, there were 59 unique sequence clusters of which the four largest sequence clusters were GPSC 4 (171; 13%), GPSC 3 (156; 12%), GPSC 7 (131; 10%), and GPSC 11 (119; 9%) (*Supplementary file 4*). There were 127 unique sequence clusters found in the Maela cohort (*Supplementary file 5*). The four largest sequence clusters were GPSC 1 (352; 13%), GPSC 28 (190; 7%), GPSC 20 (168; 6%), and GPSC 42 (123; 5%). We also looked at a subset of the Maela cohort, which included only the earliest obtained samples from unique individuals (mothers and children). This subset consisted of 762 isolates, including 380 from mother–child pairs. Isolates in this subset had the same serotypes among the most common serotypes as in the full dataset (*Supplementary file 6* and *Supplementary file 7*). Restricting this subset to mother–child paired samples only included the same

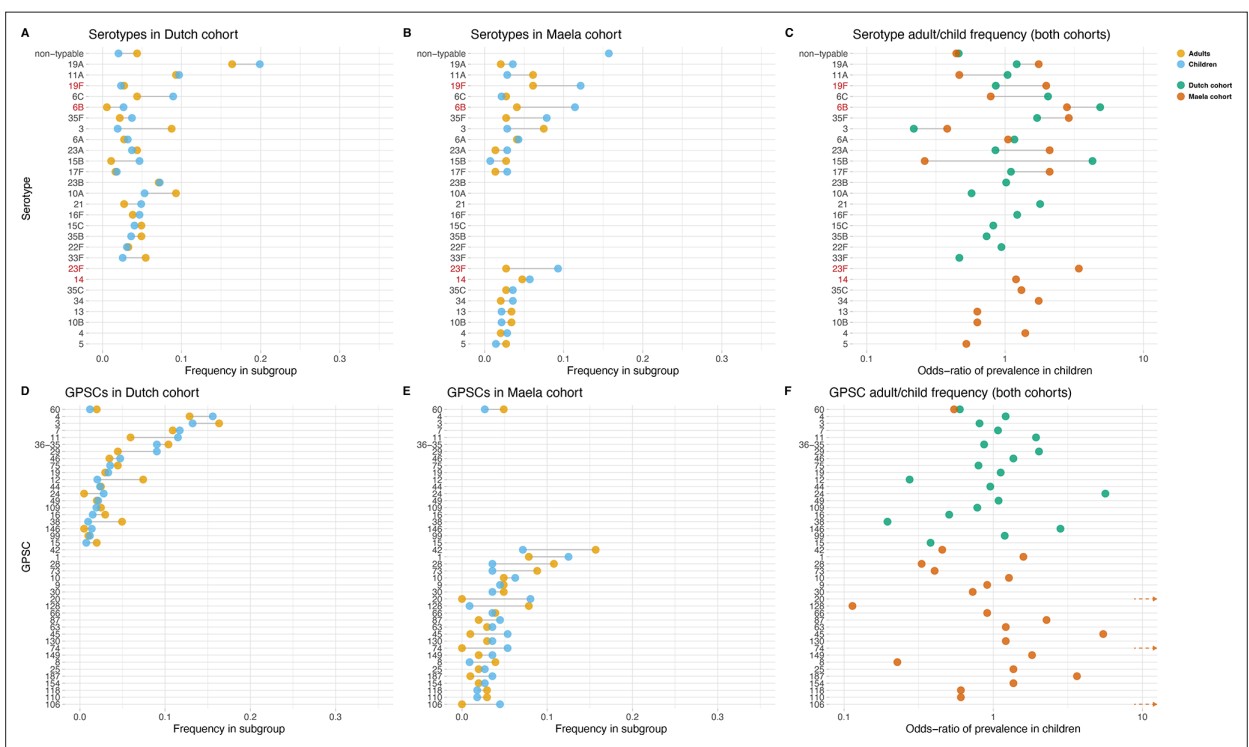

**Figure 1.** Serotype and strain (global pneumococcal sequence clusters [GPSC]) distribution by age and between cohorts. Blue dots represent frequency of serotype and strain in child carriage, yellow dots represent frequency in adult carriage. Red and green dots show odds ratio of prevalence in children in the Maela and Dutch cohorts, respectively, on a log scale for serotype. Lines show differences. Top row: dominant serotypes, ordered by presence in cohort, and internally by overall frequency. Vaccine serotypes shown in red. (**A**) Serotype frequency in the Dutch cohort. (**B**) Serotype frequency in the Maela cohort. (**C**) Comparison of adult/child log odds in each cohort for serotype. Second row: dominants strains (GPSCs), ordered by presence in cohort, and internally by overall frequency. (**D**) Strain frequency in Dutch cohort. (**E**) Strain frequency in Maela cohort. (**F**) Comparison of adult/child log odds in each cohort for strain.

The online version of this article includes the following figure supplement(s) for figure 1:

**Figure supplement 1.** Histogram for child age (in months) in (**A**) Dutch cohort (red bars) and (**B**) Maela cohort (blue bars).

serotypes and sequence clusters among the most prevalent (*Supplementary file 6* and *Supplementary file 7*).

Some serotypes exhibited a large difference in colonization frequency between the two age groups (*Figure 1*). In the Dutch cohort, serotype 6C (chi-squared test, p=0.02, not corrected for multiple testing) and serotype 15B (p=0.02) were overrepresented in children relative to adults, serotype 3

**Table 1.** Chi-squared values for serotypes in the Dutch and Maela cohorts and the age group that the serotype is affiliated with.

| Serotype | Dutch cohort | | Maela cohort | |
|---|---|---|---|---|
| | $\chi^2$ p-value | Age group | $\chi^2$ p-value | Age group |
| Non-typeable | 0.188 | Adults | $3.0 \times 10^{-4}$ | Adults |
| 19A | 0.089 | Children | 0.690 | Children |
| 11A | 0.591 | Children | 0285 | Adults |
| 19F | 1 | Adults | 0.131 | Children |
| 6C | 0.022 | Children | 1 | Adults |
| 6B | 0.099 | Children | 0.040 | Children |
| 35F | 0.279 | Children | 0.100 | Children |
| 3 | $2.5 \times 10^{-5}$ | Adults | 0.129 | Adults |
| 6A | 0.709 | Children | 1 | Children |
| 23A | 1 | Adults | - | - |
| 15B | 0.023 | Children | - | - |
| 17F | 0.943 | Children | - | - |
| 23B | 0.727 | Children | - | - |
| 10A | 0.155 | Adults | - | - |
| 15C | 1.000 | Adults | - | - |
| 35B | 0.775 | Adults | - | - |
| 22F | 1 | Adults | - | - |
| 33F | 0.132 | Adults | - | - |
| 23F | - | - | 0.040 | Children |
| 14 | - | - | 0.949 | Children |
| 35C | - | - | 0.961 | Children |
| 34 | - | - | 0.690 | Children |
| 13 | - | - | 0.756 | Adults |
| 10B | - | - | 0.756 | Adults |
| 4 | - | - | 0.966 | Children |
| 5 | - | - | 0.710 | Adults |
| 33B | - | - | 1 | Children |
| 28F | - | - | 0.652 | Children |
| 19B | - | - | 0.710 | Adults |
| 7F | - | - | 0.971 | Children |
| 20 | - | - | 0.971 | Children |
| 18C | - | - | 1 | Adults |

$\chi^2$, chi-square; -, not applicable.

was overrepresented in adults relative to children (p=2.5 × 10⁻⁵), while in the Maela cohort, serotype groups overrepresented in children were serotype 23F (chi-squared test, p=0.04) and serotype 6B (chi-squared test, p=0.04); while non-typeable serogroup was overrepresented in adults (chi-squared test, p=3.0 × 10⁻⁴) (*Table 1*). None of the 20 largest groups of sequence clusters overlapped between the cohorts. In the Dutch cohort, only GPSC 11 was significantly associated with carriage in children (chi-squared test, p=0.03, not corrected for multiple testing), while GPSC 12 (chi-squared test, p=1.2 × 10⁻⁴) and GPSC 38 (chi-squared test, p=2.1 × 10⁻⁴) were overrepresented in adults. In the Maela cohort, only sequence cluster GPSC 128 was overrepresented in children compared to adults (chi-squared test, p=0.04), while GPSC 20 (chi-squared test, p=7.0 × 10⁻³) and GPSC 74 (chi-squared test, p=0.04) were overrepresented in adults (*Table 2*).

A phylogenetic tree of pooled sequences from both cohorts, with serotype, sequence cluster, age group, and cohort for each sequence, revealed clonal discrimination between cohorts (*Figure 2*). Combined with the effects shown in *Figure 1*, this highlighted a key feature of our analysis of these datasets, which was the genetic heterogeneity between the two cohorts. Individually, each dataset clearly has strains and serotypes with strong signals of host age differences, but the overall makeup of each dataset is very different (nine common serotypes are shared, but only a single common GPSC), and where there are shared serotypes these can have different effect directions between the two cohorts.

## Host age is heritable and mostly explained by strain and serotype

To quantify the amount of variability in carriage age explained by variability in the genome, we calculated a heritability estimate ($h^2$) for each cohort. For isolates in the Dutch cohort, we did not find strong evidence that genetic variability in bacteria was related to variance in host age ($h^2$ = 0.10, 95% CI 0.00–0.69). In the Maela cohort, we found significant evidence that affinity with host age was heritable ($h^2$ = 0.56, 95% CI 0.23–0.87) and thus genetic variation in this cohort explained variation in carriage age to a greater degree. In both cohorts, pan-genomic variation could be used to predict host age to some degree of accuracy (area under the receiver-operating characteristic [ROC] curve 0.82 [Dutch cohort]; 0.91 [Maela cohort]), suggestive of some level of heritability and association of host age with strain (*Figure 3*). Prediction between cohorts using a simple linear model failed as the genetic variants chosen as predictors were not found in the other cohort – again highlighting the high level of genetic heterogeneity between cohorts.

To further investigate the association of serotype and sequence cluster to carriage age, we determined the proportion of variation in carriage age explained by serotype and sequence cluster alone. Here, we estimated $h^2_{serotype}$ = 0.07 (95% CI 0.04–0.14) and $h^2_{GPSC}$ = 0.06 (95% CI 0.03–0.13) for the Dutch cohort and $h^2_{serotype}$ = 0.11 (95% CI 0.05–0.21) and $h^2_{GPSC}$ = 0.20 (95% CI 0.12–0.31) for the Maela cohort, confirming a larger contribution of serotype and sequence cluster to carriage age heritability in sequences from the Dutch cohort. We also performed a genome-wide association analysis, but without controlling for population structure. This reveals genetic variants specific to serotype as determinants for carriage age (p-values<5.0 × 10⁻⁸) in both cohorts (*Supplementary file 8*, Dutch cohort, and *Supplementary file 9*, Maela cohort). Among the genetic variants with the lowest p-values were variants in capsule locus genes (Cps) in both cohorts. This further supports a role of strain and serotype in association with host age, but does not distinguish between the two.

## Genome-wide association analysis does not find genetic variants independent of strain

Following these observations that serotype and strain do not explain the full heritability, specifically in the Maela cohort, we performed a pathogen genome-wide association analysis to investigate whether we can detect genetic variants irrespective of the genetic background that are associated with carriage in children or adults. Though the cohorts have little genetic overlap in terms of genetic background, we would be well-powered to detect genetic variation independent of background ('locus' associations) (*Earle et al., 2016*; *Lees et al., 2016*). In the Dutch cohort, none of the unitigs, SNPs, COGs, or rare variants surpassed the threshold for multiple testing correction (*Figure 4—figure supplement 1*). The burden (sum) of rare variants in a gene for tryptophan synthase, *trpB*, approaches the multiple testing threshold, but was not significant. In the Maela cohort, unitigs in the *ugpA* gene surpassed the threshold for statistical significance (*Figure 4—figure supplement 2*), but these did

**Table 2.** Chi-squared values for strains in the Dutch and Maela cohorts and the age group that the strain is affiliated with.

| GPSC | Dutch cohort | | Maela cohort | |
|------|------------------|-----------|------------------|-----------|
| | $\chi^2$ p-value | Age group | $\chi^2$ p-value | Age group |
| 60 | 0.568 | Adults | 0.727 | Adults |
| 4 | 0.298 | Children | - | - |
| 3 | 0.392 | Adults | - | - |
| 7 | 0.858 | Children | - | - |
| 11 | 0.03 | Children | - | - |
| 35 and 36 | 0.617 | Adults | - | - |
| 29 | 0.049 | Children | - | - |
| 46 | 0.563 | Children | - | - |
| 75 | 0.666 | Adults | - | - |
| 19 | 0.978 | Children | - | - |
| 12 | $1.2 \times 10^{-4}$ | Adults | - | - |
| 44 | 1 | Adults | - | - |
| 24 | 0.094 | Children | - | - |
| 49 | 1 | Children | - | - |
| 109 | 0.817 | Adults | - | - |
| 16 | 0.249 | Adults | - | - |
| 38 | $2.1 \times 10^{-4}$ | Adults | - | - |
| 146 | 0.489 | Children | - | - |
| 99 | 1 | Children | - | - |
| 15 | 0.22 | Adults | - | - |
| 42 | - | - | 0.134 | Children |
| 1 | - | - | 0.276 | Adults |
| 28 | - | - | 0.110 | Children |
| 73 | - | - | 0.253 | Children |
| 10 | - | - | 0.777 | Adults |
| 9 | - | - | 1 | Children |
| 30 | - | - | 0.993 | Children |
| 20 | - | - | $7.0 \times 10^{-3}$ | Adults |
| 128 | - | - | 0.042 | Children |
| 66 | - | - | 1 | Children |
| 87 | - | - | 0.450 | Adults |
| 63 | - | - | 1 | Adults |
| 45 | - | - | 0.129 | Adults |
| 130 | - | - | 1 | Adults |
| 74 | - | - | 0.040 | Adults |
| 149 | - | - | 0.686 | Adults |
| 8 | - | - | 0.364 | Children |

*Table 2 continued*

| GPSC | Dutch cohort | | Maela cohort | |
| --- | --- | --- | --- | --- |
| | $\chi^2$ p-value | Age group | $\chi^2$ p-value | Age group |
| 25 | - | - | 1 | Adults |
| 187 | - | - | 0.371 | Adults |
| 154 | - | - | 1 | Adults |
| 118 | - | - | 0995 | Children |
| 110 | - | - | 0.995 | Children |
| 106 | - | - | 0.073 | Adults |

$\chi^2$, chi-square; -, not applicable; GPSC, global pneumococcal sequence clusters.

not hold after meta-analysis. After meta-analysis, there were two hits that surpassed the threshold for statistical significance (*Figure 4*).

The first is a nucleotide sequence marked by multiple unitigs of which the lowest has a p-value of $1.2 \times 10^{-9}$ (*Supplementary file 10*). This sequence does not map to the *S. pneumoniae* D39V reference sequence (*Altschul et al., 1990*) For this reason, it is not visualized on the Manhattan plot, for which unitigs were mapped to the *S. pneumoniae* D39V reference sequence (*Figure 4*). Upon inspection of the individual sequences these unitigs are called from, we find them to map in the intergenic region between open-reading frames encoding the accessory Sec-dependent serine-rich glycoprotein adhesin and a MarR-like regulator, respectively. This region contains sequences resembling transposable elements and an open-reading frame encoding a transposase. The unitigs map upstream of the start codon of the accessory Sec-dependent serine-rich glycoprotein adhesin. The sequence is present in 169 out of 1282 (13%) sequences in the Dutch cohort and in 241 out of 3085 (8%) in the Maela cohort. The sequence is present in isolates dispersed over the phylogenetic tree and associated with carriage in children (*Figure 2—figure supplement 1*). This protein is involved in adhesion to epithelial cells and biofilm formation (*Chan et al., 2020*; *Middleton et al., 2021*; *Weiser et al., 2018*). Given that this sequence lies just upstream of the start codon, it is plausible that variation of this sequence alters the expression of the Sec-dependent adhesin protein, and therefore affects carriage.

The second hit is a burden of rare variants in a gene for tryptophan synthase, *trpB*, that surpass the threshold for statistical significance at a p-value of $5.0 \times 10^{-5}$. The variants are two frameshift variants of very low frequency. These result in a predicted dysfunctional *trpB* gene in 9 out of 1282 (1%) sequences in the Dutch cohort and in 12 out of 3073 (0.4%) sequences in the Maela cohort. This association of the *trpB* gene is likely to be an artifact of low allele frequency as we estimate we are only powered to detect variation in at least 5% of isolates.

## Pilus gene presence does not determine carriage age independent of genetic background

Finally, we investigated whether pneumococcal isolates containing a pilus gene preferentially colonize children in the Dutch cohort, as has been previously described in the Maela cohort (*Binsker et al., 2020*; *Turner et al., 2012a*). This study analyzed the Maela cohort and found that 934 out of 2557 (37%) isolates in children versus 95 out of 592 (16%) isolates in adults had pilus genes present. However, this association of pilus gene presence to carriage age was dependent on lineages within the population (*Binsker et al., 2020*). In the Dutch cohort, we found no evidence that host age was dependent on pilus gene presence (22 out of 208 [10%] in adults versus 129 out of 1099 [12%] in children). This was the case whether or not the genetic background was adjusted for (p=0.35, uncorrected for population structure, and p=0.69, corrected for population structure). Based on these findings, we suggest that the previously reported pilus-IgA1 association is not a universal explanation for difference in colonization between hosts of different ages.

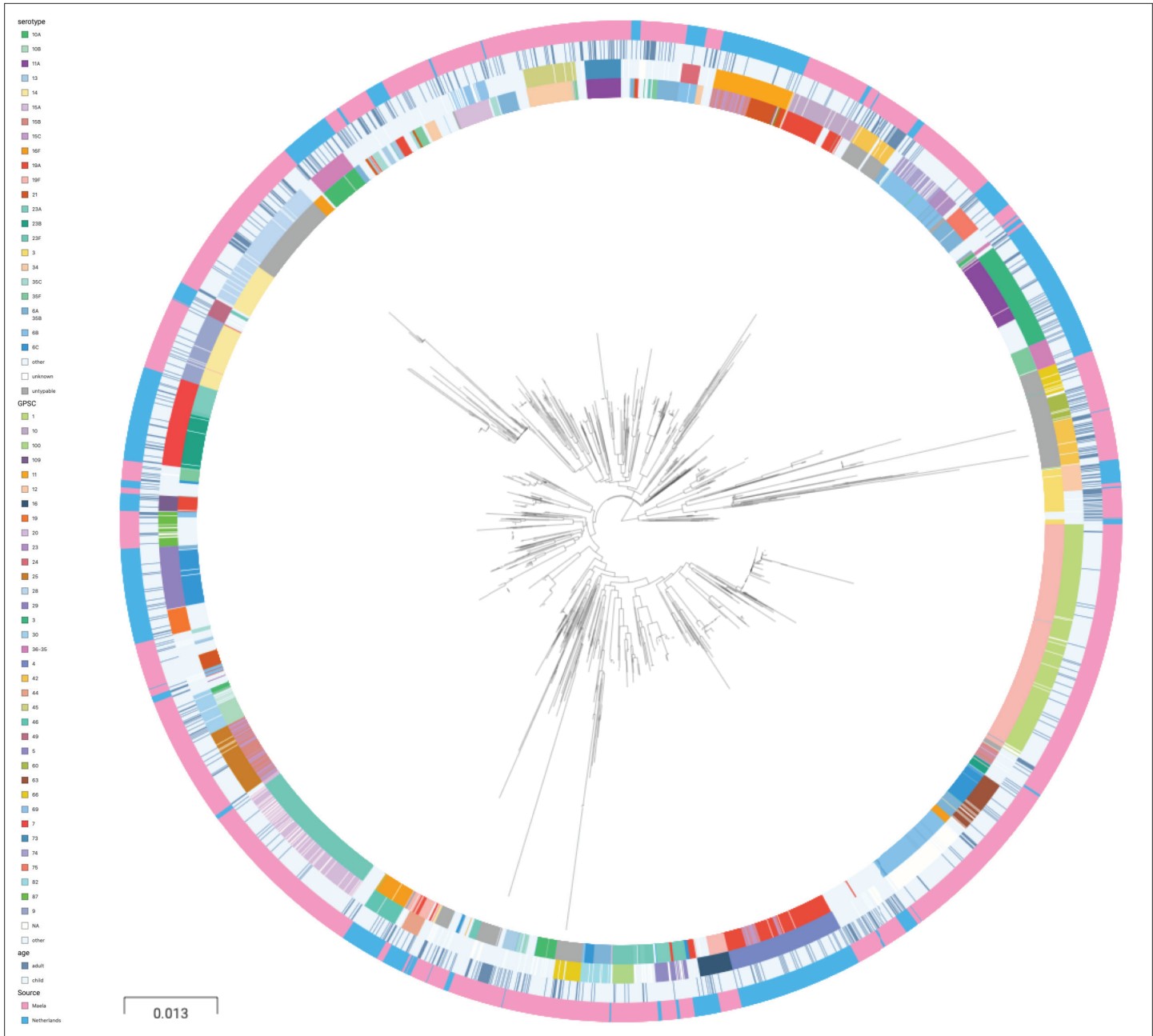

**Figure 2.** Phylogenetic tree of carriage samples from both cohorts. The rings show metadata for the samples. Depicted from inside to outside, these are serotype, sequence cluster (global pneumococcal sequence clusters [GPSC]), age, and source (Maela, Netherlands). Scale bar: 0.013 substitutions per site. An interactive version is available at here (project link available here).

The online version of this article includes the following figure supplement(s) for figure 2:

**Figure supplement 1.** Phylogenetic tree of carriage samples from both cohorts.

## Discussion

The age of the host is known to have an important effect on pneumococcal colonization (*Croucher et al., 2018*). Observational studies have demonstrated variation in serotype prevalence and carriage duration between infants and adults. Mechanistic studies in mice and humans have shown examples of differing immune responses depending both on host factors and pathogen factors. Findings from these studies include the observation that capsular polysaccharides (determinants of serotype) inhibit phagocytic clearance in animal models of upper respiratory tract colonization (*Nelson et al., 2007*).

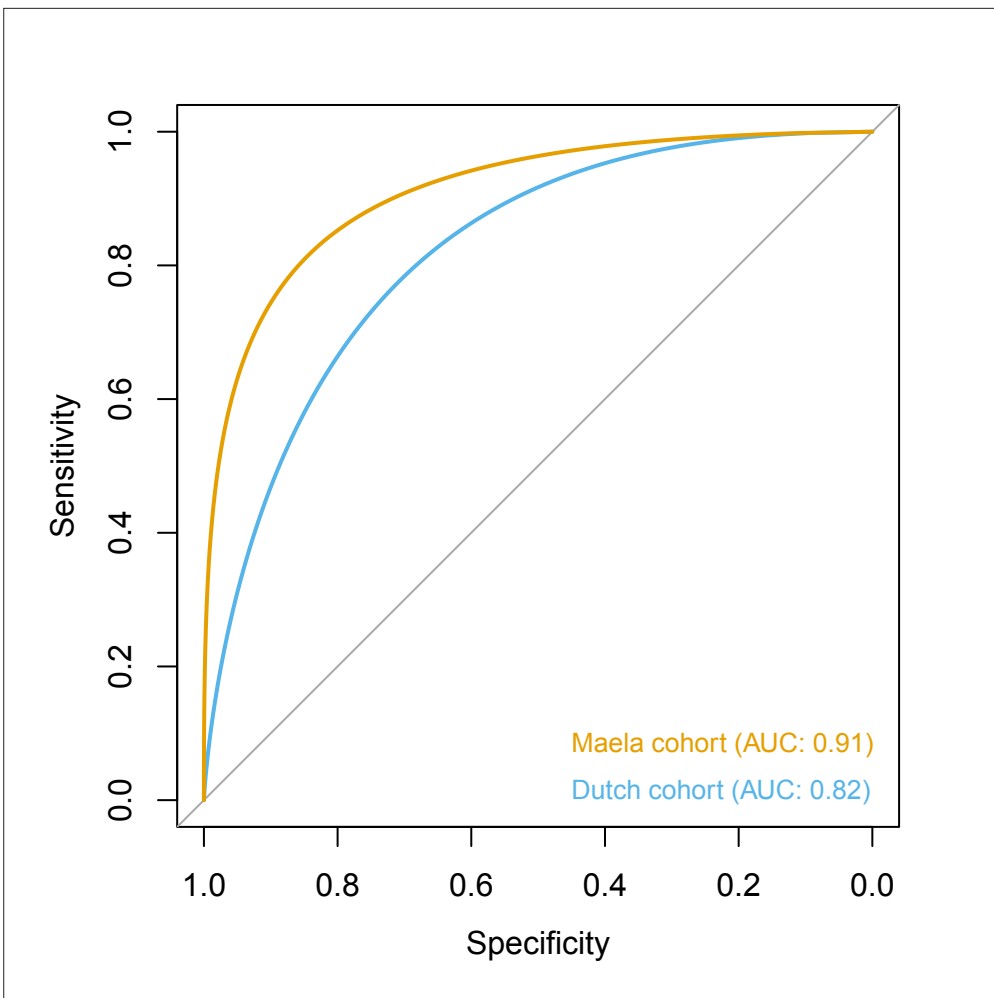

**Figure 3.** Prediction of host age from pan-genomic variation in each cohort. The smoothed receiver-operating characteristic (ROC) curve based on a linear predictor (elastic net fitted to unitigs, with strains used as folds for cross-validation) is shown. Area under the curve (AUC) is 0.5 for no predictive ability and 1 for perfect prediction.

A pneumolysin-induced IL-1 response determined colonization persistence in an age-dependent manner *Kuipers et al., 2018*; and pilus-expressing strains were found to preferentially colonize children because of immune exclusion via secretory IgA in non-naïve hosts (*Binsker et al., 2020*).

Building upon these observations, we sought to investigate and quantify the contribution of pathogen genetic variation to carriage in infant versus adult hosts using a top-down approach to systematically test the effect of pathogen genome variation on niche specificity (with age as the niche). We used whole-genome sequencing and applied statistical genetic methods to two large *S. pneumoniae* carriage cohorts. We aimed to quantify the differing patterns of serotype and strain prevalence between the two age classes during carriage and search for other genetic factors associated with host age. We show evidence that bacterial genetic variability indeed influences predilection for host age, though the effect size appears to be highly variable between populations.

Strain, or genetic background, explains 60% of the total heritability in the Dutch cohort, but only a minority in the Maela cohort. We found sequences in one region that map closely to the start codon of the accessory Sec-dependent serine-rich glycoprotein adhesin to be associated with carriage age independent of genetic background, in a meta-analysis of the two cohorts.

In previous bacterial GWAS of antimicrobial resistance (such as a single gene that causes antibiotic resistance), large monogenic effects have typically been found to have high heritabilities close to one, and the GWAS identify the causal variant precisely.(*Earle et al., 2016*; *Chewapreecha et al., 2014b*; *Lees et al., 2020*). When applied to virulence and carriage duration phenotypes, heritable effects have

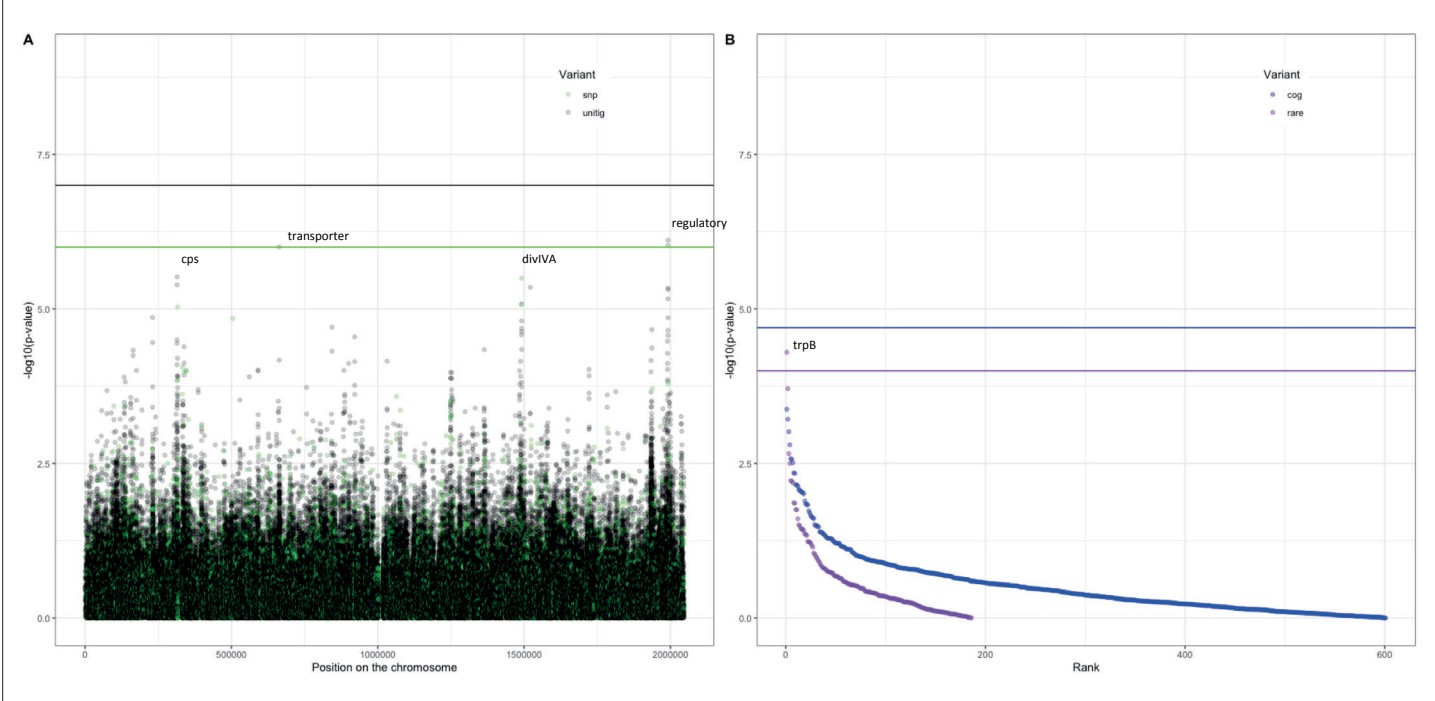

**Figure 4.** Association of variants after meta-analysis with carriage age 0–24 months. (**A**) Minus log-transformed p-value on the y-axis and position of unitig and single-nucleotide polymorphism (SNP) variants on the *S. pneumoniae* genome on the x-axis (Manhattan plot). (**B**) Minus log-transformed p-value on the y-axis and sorted lowest to highest p-value for rare variant burden in genes (purple) and clusters of orthologous genes (COGs, blue) on the x-axis.

The online version of this article includes the following figure supplement(s) for figure 4:

**Figure supplement 1.** Association of variants in the Dutch cohort with carriage age 0–24 months.

**Figure supplement 2.** Association of variants in the Maela cohort with carriage age 0–24 months.

also been found, but these only explained some of the variation in the phenotype. These appeared to be caused by many weaker effects, known as a polygenic trait, not all of which could be detected using the relatively small cohorts available (*Lees et al., 2017b*; *Lees et al., 2019a*). Polygenic effects were also seen in another study on the contribution of genetic variation to disease severity of IPD (*Cremers et al., 2019*). We found similar results for host age heritability in our two cohorts.

We could not distinguish between genetic background or serotype being the primary effect due to their correlation. We did note a difference in effect size of serotype between the two cohorts, which may make it unlikely to be the single largest effect on host age. This difference in cohorts could be explained by strain/GPSC being the main and consistent effect on host age. As strains are different between cohorts and each serotype appears in multiple strains, combining them in different amounts would create different directions of effect for serotype.

Our results are therefore suggestive of the following genetic architecture for association with host age. Primarily, whether a particular infant is colonized with a pneumococcus, when compared to an adult from the same population, is not due to the pathogen's genetics. This may be due to technical factors such as detectability related to pathogen load (which varies between adults and children, as well as between serotypes), different local forces of exposure, or other environmental or host factors such as diet (which may affect survival to *trpB*-defective pneumococci). However, some of the variation of these patterns can be explained by the pathogen's genetics. This appears largely to be driven by the fact that some strains and serotypes are more likely to be found in an infant or adult nasopharynx. In addition, there are likely to be many variants that each contribute a small amount to host age preference, but no single universally important gene or variant (a polygenic trait).

From this study, we cannot say specifically which regions of the genome contain these small effects, but it is useful to rule out recently adaptive variants with individually strong effects. We did not replicate the association of piliated genomes in infant hosts in our newly sequenced cohort, further

demonstrating important differences between populations. An important corollary of our work is on future pneumococcal vaccine optimization efforts. A promising approach for future vaccination strategies is to target the different age groups (*Colijn et al., 2020*). Whether these should consist of the dominant disease-causing serotypes overrepresented in carriage by each age group or whether there are age-specific pathogen proteins that should be included is an open question. Our study therefore suggests that targeting these age groups using serotype makeup alone would be sufficient and supports previous observational and modeling studies that advise targeting the serotype makeup in the vaccine at specific populations to maximize their effect.

Three reasons can contribute to not finding individual effects: a high proportion of the heritability being caused by lineage effects; rare locus effects that could not be detected with the current sample size; and by sampling from a cohort with vaccinated children and unvaccinated adults and comparing with a cohort of unvaccinated children and adults, we had lower power due to the reduced overlap within and between cohorts in pan-genome content. Although differences in vaccination status between cohorts is one plausible explanation for interpreting our findings, we were unable to rule out other factors, for example, a population-specific host effect, geographical differences (*Li et al., 2019*), or the broad effects of different socioeconomic status between these cohorts. Pneumococcal factors such as differences in detectability due to carriage density could also influence results.

One important difference between our study cohorts was that children from the Dutch cohort were vaccinated, while children from the Maela cohort were not. While our findings demonstrate that vaccinated versus unvaccinated children were colonized with different bacterial serotypes and different sequence clusters, we observed differences in prevalence beyond just the serotypes included in the vaccine. Another difference between the cohorts was that adults from the Dutch cohort were males and females, while adults from the Maela cohort were females only.

In summary, we found an effect of pneumococcal genetics on carriage in children versus adult hosts, which varies between cohorts, and is likely primarily driven by serotype or strain (lineage) effects rather than large population-wide effects of individual genes.

## Materials and methods
### Cohort collection
Cohorts were selected based on availability. The Dutch cohort consists of parent–child paired isolates of carriage samples from individuals obtained from three prospective carriage surveillance studies (*Spijkerman et al., 2012*; *Bosch et al., 2016*; *van Beek et al., 2017*). In these studies, carriage was assessed by conventional culture of nasopharyngeal or oropharyngeal swabs of vaccinated children (11 and 24 months of age) and their parents in 2009, 2010/2011, 2012, and 2013 (*Spijkerman et al., 2012*). All children were vaccinated with PCV-7 or PHiD-CV10 according to the Dutch national immunization program at 2, 3, 4, and 11 months of age. Vaccination status of the parents was unknown. Exclusion criteria are described elsewhere (*Spijkerman et al., 2012*; *Bosch et al., 2016*). Nasopharyngeal swabs were collected from all individuals and oropharyngeal swabs were collected from all adult subjects by trained study personnel using flexible, sterile swabs according to the standard procedures described by the World Health Organization (*O Brien et al., 2003*). After sampling, swabs were immediately placed in liquid Amies transport medium and transported to the microbiology laboratory at room temperature and cultured within 12 hr. Pneumococcal isolates were identified using conventional methods, as described previously (*Trzciński et al., 2013*). The Maela cohort consists of samples from people from a camp for displaced persons on the Thailand–Myanmar border, where monthly nasopharyngeal sampling was performed in unvaccinated children (0–24 months old) and their mothers. This cohort consists of mother–child paired samples, some of which were sampled from the same mother or child over multiple time points, and unpaired samples from mothers and children. A subselection from this cohort was made to reflect the first sampled isolates for each mother–child pair and unpaired samples to obtain isolates belonging to unique individuals. Procedures for collecting samples and generating whole-genome sequences have been previously described (*Turner et al., 2012b*; *Chewapreecha et al., 2014b*).

## Informed consent

Written informed consent was obtained from both parents of each child participant and from all adult participants. Approval for the 2009 and 2012/2013 studies in children and their parents (NL24116 and NL40288/NTR3613) was received from the National Ethics Committee in the Netherlands (CCMO and METC Noord-Holland). For the 2010/2011 study, a National Ethics Committee in The Netherlands (STEG-METC, Almere) waived the requirement for EC approval. Informed consent for the Maela cohort is described elsewhere (*Turner et al., 2012b*). Studies were conducted in accordance with the European Statements for Good Clinical Practice and the Declaration of Helsinki of the World Medical Association.

## Host age distribution in sequenced carriage cohorts

In the Dutch cohort, children had a median age of 23 months (interquartile range [IQR] 10–24 months). Adults had a median age of 35 (IQR 32–38) years. In the Maela cohort, the median age of children was 13 months (IQR 6–19 months), and for mothers (women of childbearing age) the exact age was unknown (*Figure 1—figure supplement 1*; *Turner et al., 2012b*; *Turner et al., 2013*) In the Dutch cohort, all children were vaccinated with PCV-7 or PHiD-CV10. None of the members of the Maela cohort had received PCV.

## DNA extraction and whole-genome sequencing

For the Dutch cohort, DNA extraction was performed with the Gentra Puregene Isolation Kit (QIAGEN), and quality control procedures were performed to determine yield and purity. Sequencing was performed using multiplexed libraries on the Illumina HiSeq platform to produce paired-end reads of 100 nucleotides in length (Illumina, San Diego, CA). Quality control involved analysis of contamination with Kraken (version 1.1.1)(*Page et al., 2016*), number and length of contigs, GC content, and N50 parameter. Sequences for which one or more of these quality control parameters deviated by more than 3 standard deviations from the mean were excluded. Sequences were assembled using a standard assembly pipeline (*Page et al., 2016*). Assembly statistics can be found in *Supplementary file 11*. Genome sequences were annotated with PROKKA, version 1.11 (*Seemann, 2014*). For the Maela cohort, DNA extraction, quality control, and whole-genome sequencing have been described elsewhere (*Chewapreecha et al., 2014a*). Serotypes were determined from the whole-genome sequence by in-house scripts (*Croucher et al., 2011*). Sequence clusters (strains) were defined as GPSC using PopPUNK (version 2.2.0) using a previously published reference database (*Gladstone et al., 2019*; *Lees et al., 2019b*). For 114 and 401 sequences in the Dutch and Maela cohorts, respectively, the GPSC could not be inferred due to low sequence quality.

## Sequencing characteristics and quality control

A total of 1361 bacterial isolates were sequenced as part of the Dutch cohort. During quality control, 32 sequences were excluded. Of these, 8 belonged to a different pathogen species, 9 had contamination, 14 were excluded based on the number of contigs or genome length, and 1 sequence failed annotation. For 47 sequences, host age was missing. The average length of the sequences was 2,105,305 nucleotides, with a standard deviation of 51,679 nucleotides. The mean number of contigs was 67, range 23–226. The association analyses were performed on 1282 sequences in the Dutch cohort. Of these, 1052 were isolated from children and 230 from adults. There were 3085 sequences available from the Maela cohort. Quality control for this cohort was described previously (*Chewapreecha et al., 2014a*). There were 2503 sequences isolated from children and 582 from adults. In a subset from the Maela cohort, there were 762 isolates from unique hosts, of which 380 were paired isolates (190 from children and 190 from their mothers). For the determination of the frequency and odds ratio of serotype and GPSCs in children and adults, only the first isolate from each carriage episode for each child was included in the analysis. This resulted in 964 serotypes and 799 GPSCs (165 missing) in children, and 582 serotypes and 508 GPSCs (74 missing) in adults. For adults, chi-squared tests to calculate the p-value for association between serotype and strain with age were performed in R (version 4.0.0).

## Phylogenetic tree

A core genome for sequences from both cohorts together was generated with Roary (version 3.5.0, default parameters) using a 95% sequence identity threshold (*Page et al., 2015*). A maximum

likelihood phylogeny of SNPs in the core genome of all sequenced isolates from both cohorts together was produced with IQ-TREE (version 1.6.5, including fast stochastic tree search algorithm, GTR+I+ G) assuming a general time-reversible model of nucleotide substitution with a discrete γ-distributed rate heterogeneity and the allowance of invariable sites (*Nguyen et al., 2015*).

## Heritability analysis

Based on the kinship matrix and phenotypes, a heritability estimate was performed in limix (version 3.0.4 with default parameters) for both cohorts separately (*Lippert et al., 2014*). A confidence interval around the heritability estimate was determined with Accurate LMM-based heritability Bootstrap confidence Intervals (ALBI) based on the eigenvalue decomposed distances in the kinship matrix and the heritability estimate with the gglim package (version 0.0.1) in R (version 4.0.0) (*Schweiger et al., 2018*). To estimate the proportion of heritability attributable to serotype or strain alone, we calculated the heritability with limix, based on a kinship matrix treating serotypes or strains as genetic variants (*Lees et al., 2017b*; *Lees et al., 2018*). Again, a confidence interval around the heritability estimate was determined with ALBI (*Schweiger et al., 2018*). The code used to perform these analyses is available at https://github.com/philipkremer123/carriage_pneumo_heritability (*Kremer, 2022a* copy archived at swh:1:rev:73c4fa5c8d24d76945308b-2616fbb5572d0c39b4) and https://github.com/johnlees/carriage-age-plots, (*Kremer, 2022b* copy archived at swh:1:rev:f9477d6b8382fee6926be7fd29b99afc15873fe8).

## Determining bacterial genetic variation: Unitigs, SNPs, and COGs

Using the whole-genome sequence reads from both cohorts, we called SNPs, small insertions and deletions, and SNPs clustered as rare variants (deleterious variants at an allele frequency < 0.01) based on the *S. pneumoniae* D39V reference (CP027540) sequence using the Snippy pipeline (version 4.4.0, default parameters). We determined nonredundant sequence elements (unitigs) from assembled sequences in the Dutch cohort by counting nodes on compacted De Bruijn graphs with Unitig-counter (version 1.0.5, default minimum k-mer length of 31) (*Jaillard et al., 2018*). These unitigs were called in an indexed set of sequences from the Maela cohort with Unitig-caller (version 1.0.0, default parameters) (*Lees et al., 2020*). This gave us the distribution of sequences from both cohorts with consistent k-mer definitions, making it possible to run predictive models across cohorts. The same Roary run as was used to generate the core-genome alignment was used to extract accessory COGs (*Page et al., 2015*).

There were 9,966,794 unitigs counted from combined sequences in the Dutch cohort. Of these, 303,901 passed a minor allele frequency (MAF – the frequency of isolates a genetic sequence, or allele, is identified in) of 0.05 filter and had association testing performed. The 9,966,794 unitigs from the Dutch cohort were called in sequences from the Maela cohort to obtain 726,040 unitigs. Association testing in this group was done for 323,112 unitigs that were present at MAF 0.05 or more. Meta-analysis was performed on 251,733 overlapping unitigs. There were 313,143 SNPs called from sequences in the Dutch cohort, of which 43,556 passed MAF filtering. For the Maela cohort, 382,230 SNPs were called and 53,553 passed the MAF filter. For meta-analysis, 20,118 SNPs had overlapping positions and were included. There were 1997 rare variants called in the Dutch cohort, which were burdened in 538 genes. For the Maela cohort, these numbers were 1997 and 423. Together, 186 genes were included in the meta-analysis. Lastly, 2348 COGs were analyzed in the Dutch cohort and 4678 in the Maela cohort. In the meta-analysis, there were 627 overlapping COGs.

## Genome-wide association study

The association analysis for SNPs, unitigs, rare variants, and COGs was run as a linear mixed model in Pyseer (version 1.1.1), with a minimum MAF of 0.05 (*Lees et al., 2018*). To correct for population structure, the model included a kinship matrix as covariates, which was calculated from the midpoint rooted phylogenetic tree. An association analysis not corrected for population structure was run with unitigs as sequence elements using a simple fixed-effects model in Pyseer. Rare variants were clustered in their corresponding gene and analyzed in a burden test. Meta-analysis was performed on summary statistics from the Pyseer results files with METAL (version released on August 28, 2018, default parameters) for each variant (*Willer et al., 2010*). A threshold for association of the phenotype with meta-analyzed variants was determined using a Bonferroni correction with alpha < 0.05 and the

number of independent tests in the Dutch cohort, giving $p<1.0 \times 10^{-7}$ for unitigs, $p<1.0 \times 10^{-6}$ for SNPs, $p<2.0 \times 10^{-5}$ for COGs, and $p<1.0 \times 10^{-4}$ for rare variants. Unitigs were mapped to the *S. pneumoniae* D39V reference genome with bowtie-2 (version 2.2.3, with equal quality values and length of seed substrings 7 nucleotides). In accordance with the study populations in both cohorts, the phenotype was dichotomized as host age 0–24 months versus adult age. Manhattan plots were generated in R (version 3.5.1) with the package ggplot2 (version 3.1.0). Presence or absence of pilus genes was detected by nucleotide BLAST (version 2.6.0, default parameters) analysis. Pilus gene presence association to carriage age was calculated with a likelihood ratio test in Pyseer (version 1.1.1), corrected for population structure by including a kinship matrix as covariates.

The prediction analysis used the elastic net mode of Pyseer. This fitted an elastic net model with a default mixing parameter (0.0069 L1/L2) to the unitigs counted in each cohort using the strains from PopPUNK as folds to try and reduce overfitting (*Lees et al., 2020*). ROC curves for each cohort were drawn using the linear link output, with the R package pROC (version 1.16.2) using smoothing. To test inter-cohort prediction, the called unitigs from the other cohorts were used as predictors with the model from the opposing cohort.

## Acknowledgements

We thank Dr. Nicholas Croucher from Imperial College London for commenting on the manuscript. This work was supported by grants from the European Research Council (ERC Starting Grant, proposal/contract 281156; https://erc.europa.eu) and the Netherlands Organization for Health Research and Development (ZonMw; NWO-Vici grant, proposal/contract 91819627; https://www.zonmw.nl/nl/), both to DvdB. Work at the Wellcome Trust Sanger Institute was supported by Wellcome Trust core funding (098051; https://wellcome.ac.uk). JAL was funded by Wellcome (219699) and received support from the Medical Research Council (grant number MR/R015600/1). This award is jointly funded by the UK Medical Research Council (MRC) and the UK Department for International Development (DFID) under the MRC/DFID Concordat agreement and is also part of the EDCTP2 program supported by the European Union. PT was funded in part by the Wellcome Trust (grant number 083735/Z/07/Z). The Netherlands Reference Laboratory for Bacterial Meningitis was supported by the National Institute for Health and Environmental Protection, Bilthoven (https://www.rivm.nl/). For the purpose of open access, the authors have applied a CC BY public copyright licence to any Author Accepted Manuscript version arising from this submission. The funders had no role in study design, data collection and analysis, decision to publish, or preparation of the manuscript.

## Additional information

### Funding

| Funder | Grant reference number | Author |
| --- | --- | --- |
| European Research Council | 281156 | Diederik van de Beek |
| ZonMw | 91819627 | Diederik van de Beek |
| Wellcome Trust | 219699 | John A Lees |
| Wellcome Trust | 083735/Z/07/Z | Paul Turner |
| Rijksinstituut voor Volksgezondheid en Milieu | | Arie van der Ende |

The funders had no role in study design, data collection and interpretation, or the decision to submit the work for publication. For the purpose of Open Access, the authors have applied a CC BY public copyright license to any Author Accepted Manuscript version arising from this submission.

### Author contributions

Philip HC Kremer, Conceptualization, Formal analysis, Methodology, Writing - original draft, Project administration, Writing – review and editing; Bart Ferwerda, Formal analysis, Supervision,

Methodology, Writing – review and editing; Hester J Bootsma, Nienke Y Rots, Alienke J Wijmenga-Monsuur, Elisabeth AM Sanders, Krzysztof Trzciński, Anne L Wyllie, Paul Turner, Data curation, Writing – review and editing; Arie van der Ende, Supervision, Methodology, Writing – review and editing; Matthijs C Brouwer, Supervision, Writing – review and editing; Stephen D Bentley, Diederik van de Beek, Supervision, Funding acquisition, Methodology, Writing – review and editing; John A Lees, Conceptualization, Formal analysis, Supervision, Methodology, Writing – review and editing

### Author ORCIDs
Philip HC Kremer (ID) http://orcid.org/0000-0003-0483-841X
Alienke J Wijmenga-Monsuur (ID) http://orcid.org/0000-0001-5663-860X
Anne L Wyllie (ID) http://orcid.org/0000-0001-6015-0279
Paul Turner (ID) http://orcid.org/0000-0002-1013-7815
John A Lees (ID) http://orcid.org/0000-0001-5360-1254

### Ethics
in children and their parents (NL24116 and NL40288/NTR3613) were received from the National Ethics Committee in the Netherlands (CCMO and METC Noord-Holland). For the 2010/2011 study, a National Ethics Committee in The Netherlands (STEG-METC, Almere) waived the requirement for EC approval. Informed consent for the Maela cohort was described elsewhere. Studies were conducted in accordance with the European Statements for Good Clinical Practice and the Declaration of Helsinki of the World Medical Association.

### Decision letter and Author response
Decision letter https://doi.org/10.7554/eLife.69244.sa1
Author response https://doi.org/10.7554/eLife.69244.sa2

---

## Additional files

### Supplementary files
• Supplementary file 1. Serotypes in the Dutch cohort, and the number of samples isolated from child or adult.

• Supplementary file 2. Number of samples for each of the vaccine serotypes found in the Dutch cohort.

• Supplementary file 3. Serotypes in the Maela cohort, and the number of samples isolated from child or mother.

• Supplementary file 4. Strains (global pneumococcal sequence clusters [GPSCs]) in the Dutch cohort, and the number of samples isolated from child or adult.

• Supplementary file 5. Strains (global pneumococcal sequence clusters [GPSCs]) in the Maela cohort, and the number of samples isolated from child or mother.

• Supplementary file 6. Serotypes and strains (global pneumococcal sequence clusters [GPSCs]) in the subset of the Maela cohort with unique samples only, and the number of samples isolated from child or mother for each, including percentages.

• Supplementary file 7. Serotypes and strains (global pneumococcal sequence clusters [GPSCs]) in the subset of the Maela cohort with unique paired (mother–child) samples only, and the number of samples isolated from child or mother for each, including percentages.

• Supplementary file 8. Unitigs associated with carriage age in the Dutch cohort when not corrected for population structure of the bacterial population (lrt-p-value). The other columns provide parameters of the regression line for the unitig. The final column (annotation) provides the location of the unitig in the Streptococcus_pneumoniae_D39V genome.

• Supplementary file 9. Unitigs associated with carriage age in the Maela cohort when not corrected for population structure of the bacterial population (lrt-p-value). The other columns provide parameters of the regression line for the unitig. The final column (annotation) provides the location of the unitig in the Streptococcus_pneumoniae_D39V genome.

• Supplementary file 10. Unitigs represent the top hits for carriage age after meta-analysis of both cohorts. These unitigs are not found in any currently available reference genome, but are found to be upstream of an accessory Sec-dependent serine-rich glycoprotein adhesin in a subset of samples from these cohorts.

- Supplementary file 11. Statistics on the assembly of the sequences from the Dutch cohort.
- Supplementary file 12. Sample name, sample accession, lane name, and lane accession in the European Nucleotide Archive for the sequences from the Dutch cohort.
- Supplementary file 13. Sample name, sample accession, lane name, and lane accession in the European Nucleotide Archive for the sequences from the Maela cohort.
- Transparent reporting form

### Data availability

Fastq sequences of bacterial isolates from the Dutch cohort were deposited in the European Nucleotide Archive (ENA, study and accession numbers in Supplementary file 12). Sequences of bacterial isolates in the Maela cohort are available at ENA under study numbers ERP000435, ERP000483, ERP000485, ERP000487, ERP000598 and ERP000599 (Supplementary file 13). Summary statistics for the results from the genome wide association studies can be found at https://figshare.com/articles/dataset/S_pneumoniae_carriage_GWAS/14431313.

The following previously published dataset was used:

| Author(s) | Year | Dataset title | Dataset URL | Database and Identifier |
|---|---|---|---|---|
| Kremer PHC | 2020 | *Streptococcus pneumoniae* evolution and population structure during longitudinal sampling in a defined human population | https://www.ebi.ac.uk/ena/browser/view/PRJEB2357 | ENA, PRJEB2357 |

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
