## [Editor Report]

Strain variability in bacterial infections is a confounding factor in the treatment and prevention of the associated diseases. Pneumococcal disease is widespread, and the current vaccine targets only a subset of circulating strains, with disease and vaccine efficacy likely varying with the age of the host. Using two large databases of pneumococcal genomes, this study explores the associations between genomic factors and the age of the human host. Ultimately, these data and related studies will establish whether and how vaccines should be differentially designed for children and the elderly. This work will be of interest to those working in bacterial infections and host–pathogen genomics.

---

## [Decision Letter]

**Decision letter after peer review:**

Thank you for submitting your article "Pneumococcal genetic variability influences age-dependent bacterial carriage" for consideration by *eLife*. Your article has been reviewed by 3 peer reviewers, and the evaluation has been overseen by Bavesh Kana as the Senior Editor. The following individual involved in review of your submission has agreed to reveal their identity: Amelieke Cremers (Reviewer #2).

Essential revisions:

Major Comments:

1. The Maela set is that these are mother-child samples. These seem like excellent sets to establish genomic features associated with age. Can the data be plotted for paired mother and child? At least for the serotype and sequence types that appear age-related.

2. The work is performed to inform future pneumococcal vaccine formulations. The authors looked for genome-wide pneumococcal traits that made them more likely to colonize a human host of particular age category. Although the bioinformatics analyses are extensive and of high quality, the sampling strategy needs clarification before the conclusions can be evaluated.

3. The manuscript may benefit from a hypothesis on what was expected. Were parents expected to acquire pneumococcal carriage from sources outside of their children? Or did the authors expect differences in viability or detectability of particular pneumococcal variants in adult carriage – i.e. a subset? What question does this study set up answer.

4. Is it possible that the major finding is explained by sampling rather than a biological phenomenon? The eigenvalue decomposed distance in the kinship matrix is a measure to express the distance between isolates within a group. And the authors used this measure to signal overrepresentation of (undefined) genomic variants in either category. For infants, in the preceding serotype-based comparison only a first isolate of a carriage episode was included, discarding 75% of the (duplicate?) isolates. It is unclear whether repetitive isolates from adults and infants were included in the 'heritability analyses'. Could the authors clarify what samples of genomes were included in the kinship matrices, and how such choices may affect the representativity of results?

5. Presentation of the results could be improved. While it is understandable that the sequencing data need to be compressed to a presentable format, essential information needs to be logically displayed for the sake of readers' understanding. As examples, the first section of the result section mentioned total numbers of isolates and serotypes from each cohort, but did not say how many of them were from children/adults. Figure 1 does have age information, but it is difficult to evaluate due to data transformation. Sequence clusters were mentioned without elaboration on what they mean. This style of data presentation may be readily comprehensive for sequencing gurus, but is hard to digest to the experimentalists like myself.

Other comments that should be addressed:

1. Supplementary Tables S8 and 9 would benefit from an annotations (or representative gene ID) associated with each of the regions.

2. Paragraph starting line 214 and Figure 4: Given that genes were mapped to D39V, it would be very useful to provide an ID. Similarly, TableS10 has a column for Marker Name, but the IDs are not present only sequence segments.

3. The description of the performed 'heritability analyses' requires more detail to enable the readership to repeat the analyses.

4. The title of the manuscript may be reconsidered given the cross-sectional nature of the study performed.

5. The discussion should contain a comparison to previous studies that performed pneumococcal GWAS on host age (f.e. doi.org/10.1093/cid/ciy417). The observed geographical differences in pneumococcal elements being associated with phenotype could also be related to previous studies reporting on this phenomenon (f.e. doi: 10.1038/s41467-018-07997-y).

6. The authors may also want to discuss potential biases or alternative explanations for observed differences between infants and adults (f.e. detectability – while pneumococcal carriage density is lower in adults compared to children, serotype 3 generally grows with high load and is relatively more likely to be detected if present. Viability – Survival of trpB defective pneumococci may depend on age-dependent host factors like diet).

7. Why do the authors present epidemiological arguments to prefer either serotypes or proteins as vaccine targets, while serotypes strongly co-occur with specific proteins across the pneumococcal population that could evenly be targeted? If they choose to do so, however, the report would benefit from concrete numbers to support the claim that based on this study serotypes would be a preferential vaccine target. Which vaccine-formulations would the authors recommend for the infant and parent populations studied here?

---

## [Author Response]

Essential revisions:Major Comments:1. The Maela set is that these are mother-child samples. These seem like excellent sets to establish genomic features associated with age. Can the data be plotted for paired mother and child? At least for the serotype and sequence types that appear age-related.

Our understanding is that the suggestion here is to look at mother-child pairs where direct transmission had been identified. Indeed, this would be an excellent set to identify age-related features, as genetic background would no longer be a major confounder, and variants occurring across the bottleneck over multiple events could be pooled (as is often done in within-host studies), especially if direction of loss/gain could also be inferred.

However, reconstructing transmission pairs with single colony-pick samples alone has proved to be highly uncertain, such that we cannot infer them in this dataset (despite trying various methods previously). They are also relatively rare events to observe – many of the mother/child series have clearly non-overlapping carriage episodes.

A newer analysis of swabs from this cohort was designed specifically to detect these events by sequencing within-host diversity to help improve power to reconstruct these events (https://doi.org/10.1101/2022.02.20.480002), and targeted those swabs from timepoints likely to be involved in transmission events. Despite this, those authors found only 40 direct mother-infant transmissions. This is a low number to pool variants, and would be the best case scenario here. So, there is insufficient power in this study to (a) find these pairs and (b) find common variation between them. We also note that the genome samples do not overlap with this newer study so we cannot simply lift their analysis over.

The advantage of doing a GWAS over well-controlled pairs is that one can pool samples from a variety cases and controls, and after accounting for biases as much as possible, allow the larger sample sizes to give the power to find associations.

So, while we welcome this suggestion, due to sampling design of the cohort we do not think it possible here. However, we have added this suggestion while explaining our study rationale in the introduction, and how it differs from what we did, and why it is difficult to do with these samples.

2. The work is performed to inform future pneumococcal vaccine formulations. The authors looked for genome-wide pneumococcal traits that made them more likely to colonize a human host of particular age category. Although the bioinformatics analyses are extensive and of high quality, the sampling strategy needs clarification before the conclusions can be evaluated.

We have made significant edits to the introduction to better motivate and describe our study, also to address point three below. Our sampling strategy is basically an opportunistic one: we collected and collated as many population-matched adult and child samples as possible to maximize statistical power, and use established techniques from GWAS to account for genetic background confounding results (as well as other covariates, where appropriate).

Furthermore, we have clarified the sampling strategy in the methods section of the manuscript. On page 16, line 382 “Cohorts were selected based on availability.”. On line 677 we added “parent-child paired isolates of”. On page 16, line 395 – 396:

“This cohort consists of mother – child paired samples, some of which were sampled from the same mother or child over multiple time points, and unpaired samples from mothers and children. A subset from this cohort was made to reflect the first sampled isolates for each mother – child pair and unpaired samples, to obtain isolates belonging to unique individuals.” And in the section on “*Sequencing characteristics and quality control*”, page 17, line 441 – 442: “In a subset from the Maela cohort, there were 762 isolates from unique hosts, of which 380 were paired isolates (190 from children and 190 from their mothers).”

Isolates in this subset are listed in Supplementary File 4 and this table is now referenced clearly in the Results section.

We also felt, while addressing these concerns, we had given too much weight to implications on vaccine design, where fully modelling these effects are outside the scope of this study. We have therefore reworded some of the introduction and abstract to be more specific about the contribution this study actually makes in vaccine design, and what further work would be required before recommendations could be made.

3. The manuscript may benefit from a hypothesis on what was expected. Were parents expected to acquire pneumococcal carriage from sources outside of their children? Or did the authors expect differences in viability or detectability of particular pneumococcal variants in adult carriage – i.e. a subset? What question does this study set up answer.

Reading our introduction again we can see that the motivation for a genome-wide association study was not necessarily clear, and we appreciate the suggestion of framing in terms of hypotheses. We do not look for direct transmission events between adults and children, as noted above (and in answer to the question, broadly we expect parents will acquire *S. pneumoniae* both from their children and from others in the community). Previous studies have observed differences in carriage duration, shedding and strain makeup between adults and children, which leads to the question of which factors cause these differences. If there are genetic factors, these should be accounted for in any models of carriage (particularly those which underlie transmission models used to propose vaccine design).

We therefore test for three things using our ‘opportunistic’ sample collection:

1. Whether carriage rates of individual strains or serotypes varies substantially between infants and adults in the same contact networks.

2. Whether this variation is attributable at least in part due to pathogen genetic adaptation to either the infant or adult nasopharynx, which are immunologically different niches.

3. Finally, whether this adaptation is due both to serotype and genetic background, and that some of the genetic effects are resolvable to individual genes, alleles or regulatory variants which arise on multiple different genetic backgrounds due to a selective advantage.

If we find clear associations, this would support proposals for age-specific vaccine design, and may also suggest specific protein components which more broadly suppress carriage in the target age group than multivalent PCV alone.

Please see the rewritten introduction for full details.

4. Is it possible that the major finding is explained by sampling rather than a biological phenomenon? The eigenvalue decomposed distance in the kinship matrix is a measure to express the distance between isolates within a group. And the authors used this measure to signal overrepresentation of (undefined) genomic variants in either category. For infants, in the preceding serotype-based comparison only a first isolate of a carriage episode was included, discarding 75% of the (duplicate?) isolates. It is unclear whether repetitive isolates from adults and infants were included in the 'heritability analyses'. Could the authors clarify what samples of genomes were included in the kinship matrices, and how such choices may affect the representativity of results?

Thank you for this important remark. Only the Maela cohort had multiple samples isolated longitudinally for individual children, but using estimates from a previous study we were able to find those samples which are from the same carriage episode. These samples were removed so only one sample is taken from any given carriage episode. The heritability analyses for this cohort have now been repeated with unique samples only. This did not significantly change our conclusions. For the genome-wide association study, we believe it was best to retain all samples, because in this case the kinship matrix adjusts for the repeated sampling, but retains any diversity accumulated across the carriage episode. Furthermore, the meta-analysis with the second cohort will remove any spurious findings. We have now mentioned more clearly what samples were analyzed for each experiment, e.g. page 7, line 173 – 174:

“We also looked at a subset of the Maela cohort, which included only the earliest obtained samples from unique individuals (mothers and children). This subset consisted of 762 isolates, including 380 from mother – child pairs.”

5. Presentation of the results could be improved. While it is understandable that the sequencing data need to be compressed to a presentable format, essential information needs to be logically displayed for the sake of readers' understanding. As examples, the first section of the result section mentioned total numbers of isolates and serotypes from each cohort, but did not say how many of them were from children/adults. Figure 1 does have age information, but it is difficult to evaluate due to data transformation. Sequence clusters were mentioned without elaboration on what they mean. This style of data presentation may be readily comprehensive for sequencing gurus, but is hard to digest to the experimentalists like myself.

Thank you for pointing this out. We have gone through the results, particularly the first section, trying harder to think about viewing them from the perspectives of experts from other disciplines.

The section titled “Serotypes and strains are variably carried between age groups, and between cohorts” (page 7, line 162) was intended as a global summary of population composition for both cohorts. There are many serotypes and strains, which are difficult to summarize in text and further adding the distribution over age categories would add more to this (we have tried this presentation, but it is poor in prose). To compare between cohorts and across strains/serotypes the transform from counts to frequencies (as in figure 1) is necessary to see the information on the correct scale. However, we appreciate that a lack of age split is not helpful, and have added this essential information to the start of this section. We have also added the requested age splits of each serotype as a table, where they can be more easily read.

The reference to Figure 1 here was incorrect and has been removed. Also, the section has been simplified and information has been moved to supplementary tables to increase readability. Figure 1 is now referenced in the 3rd paragraph of the Results section (page 8, line 190), as this section presents the differences in colonization frequency in relation to age groups.

Other comments that should be addressed:1. Supplementary Tables S8 and 9 would benefit from an annotations (or representative gene ID) associated with each of the regions.

These have now been added.

2. Paragraph starting line 214 and Figure 4: Given that genes were mapped to D39V, it would be very useful to provide an ID. Similarly, TableS10 has a column for Marker Name, but the IDs are not present only sequence segments.

In Supplementary File 10 we have inserted a first column named ‘Sequence’. This column now provides the sequence segments. The second column, ‘MarkerName’ now has ‘intergenic’ for each segment sequence as these sequences do not map to an open reading frame (or any other sequence) in the D39V reference (or any other available reference), and therefore have not been allocated a gene ID. These sequences do map to the intergenic region between open reading frames encoding the accessory Sec-dependent serine-rich glycoprotein adhesen and a MarR like regulator in a subset of pneumococcal isolates from this study (but none of the available reference sequences).

3. The description of the performed 'heritability analyses' requires more detail to enable the readership to repeat the analyses.

The code used to perform the analyses can now be found at https://github.com/philipkremer123/carriage_pneumo_heritability/blob/main/code (page 18, line 474)

4. The title of the manuscript may be reconsidered given the cross-sectional nature of the study performed.

We have changed the title of the manuscript to “pneumococcal genetic variability in age-dependent bacterial carriage”.

5. The discussion should contain a comparison to previous studies that performed pneumococcal GWAS on host age (f.e. doi.org/10.1093/cid/ciy417). The observed geographical differences in pneumococcal elements being associated with phenotype could also be related to previous studies reporting on this phenomenon (f.e. doi: 10.1038/s41467-018-07997-y).

Thank you for these suggestions. We have added a comparison to the reference on genetic variation to IPD disease severity to the discussion (page 13, line 324 – 325 “Polygenic effects were also seen in another study on the contribution of genetic variation to disease severity of invasive pneumococcal disease”). Geographical differences, with a reference to the 2019 Nature Communications paper were also added (page 14, line 363: “geographical differences”)

6. The authors may also want to discuss potential biases or alternative explanations for observed differences between infants and adults (f.e. detectability – while pneumococcal carriage density is lower in adults compared to children, serotype 3 generally grows with high load and is relatively more likely to be detected if present. Viability – Survival of trpB defective pneumococci may depend on age-dependent host factors like diet).

We have added wording to the discussion on differences in carriage density, on page 14, line 364 – 365: “Pneumococcal factors such as differences in detectability due to carriage density could also influence results”. Thanks for these interesting examples, which we have also included in the text.

7. Why do the authors present epidemiological arguments to prefer either serotypes or proteins as vaccine targets, while serotypes strongly co-occur with specific proteins across the pneumococcal population that could evenly be targeted? If they choose to do so, however, the report would benefit from concrete numbers to support the claim that based on this study serotypes would be a preferential vaccine target. Which vaccine-formulations would the authors recommend for the infant and parent populations studied here?

Regarding co-occurrence of serotypes and proteins: previous work has shown that many proteins are present in every member of the population, and that proteins which are variably present (and often associated with specific serotypes) are selected to favor a frequency which is independent of vaccine driven serotype dynamics [Azarian et al., PLoS Biol. 2020; Corander et al., Nat Eco Evol 2017.]. Protein targets may therefore be desirable as they could target the entire population in a serotype-independent manner [Colijn et al., Nat Microbiol 2020]. Modifying the carrier protein used in the conjugate vaccine is one way to include a protein target using existing vaccine technology. We therefore wished to explore whether an alternative carrier protein found more commonly in infant carriage would be viable (given that reducing infant carriage has the greatest effect on reducing burden of disease), but we did not find a suitable candidate from our hypothesis-generating study.

While we appreciate the interesting suggestion to propose a particular formulation, pneumococcal population and vaccine dynamics are too complex for us to model in the scope of our paper. This is due to needing to model frequency dependent selection, vaccines which only target part of the pneumococcal population, and challenges quantifying disease burden from noisy observational study. This has previously been done by Colijn et al., Nat Microbiol 2020, who modelled this using optimization criteria, and included using part of the pilus as the carrier protein, and administering complementary adult vaccines.

Rather than simply citing this previous work on vaccine modelling, we have now added a more in-depth description of its relation to our work.